# LEARNING TO REASON ABOUT AND TO ACT ON PHYSICAL CASCADING EVENTS [*]

**Yuval Atzmon**[†]**, Eli A. Meirom**[†]**, Shie Mannor, Gal Chechik**
NVIDIA Research, Israel
`{yatzmon,emeirom}@nvidia.com`

## ABSTRACT

Reasoning and interacting with dynamic environments is a fundamental problem in AI, but it becomes extremely challenging when actions can trigger cascades of cross-dependant events. We introduce a new learning setup called *Cascade* where an agent is shown a video of a simulated physical dynamic scene, and is asked to *intervene* and trigger a cascade of events, such that the system reaches a "counterfactual" goal. For instance, the agent may be asked to "*Make the blue ball hit the red one, by pushing the green ball*". The problem is very challenging because agent interventions are from a continuous space, and cascades of events make the dynamics highly non-linear.

We combine semantic tree search with an event-driven forward model and devise an algorithm that learns to search in semantic trees in continuous spaces. We demonstrate that our approach learns to effectively follow instructions to intervene in previously unseen complex scenes. Interestingly, it can use the observed cascade of events to reason about alternative counterfactual outcomes.

## 1 INTRODUCTION

Cascades of events are wide-spread phenomena found in dynamical systems from biology (gene expression) and physics (meteorology) to chemistry and economics (supply chains). People can reason about such cascades and plan how to intervene to achieve a desired outcome of a system. This builds on several human capacities: inferring causal relations, reasoning in a counterfactual way about possible alternative dynamics of a system, and describing the goal in natural, semantic, language. Cascading dynamical systems are therefore a fantastic test bed for studying complex reasoning. But, how can we train computational agents to reason and intervene in a cascading dynamical system?

Here we address this question in a simulated dynamical system of a physical world, where object interactions form a cascade of events. We describe a new supervised learning setup, called *Cascade* (Figure 1). In this setup, an agent observes a cascade of events in a system of moving and static objects. It is then provided with a desired goal for the system described in semantic terms. The agent may intervene and manipulate one object to achieve the goal.

More concretely, we consider the following learning setup. At training time, the agent is given the initial conditions and object trajectories of an observed cascade, a semantic goal, and the initial conditions and trajectories of one possible solution. At test time, we sample a *new* unseen system and goal. The agent only has access to the goal and the "observed" cascade, and it should predict how to change the initial conditions to meet the goal.

This problem is very challenging for several reasons. First, the intervention space is continuous, but is fragmented into many regions, each yielding a different outcome. The region boundaries are hard and with no clear submodularity, as a slight change in movement direction can yield a qualitatively different final outcome. This "butterfly effect" is indeed common in cascading systems. As a result, it is hard to apply standard planning algorithms or gradient-based methods.

To further illustrate the difficulty of this task, consider how hard it is to even apply a "brute-force" approach. Such an approach would actually require access to a simulator of the system to test

---

[*]A detailed version of this work is described under: https://arxiv.org/abs/2202.01108
[†]Equal contribution.

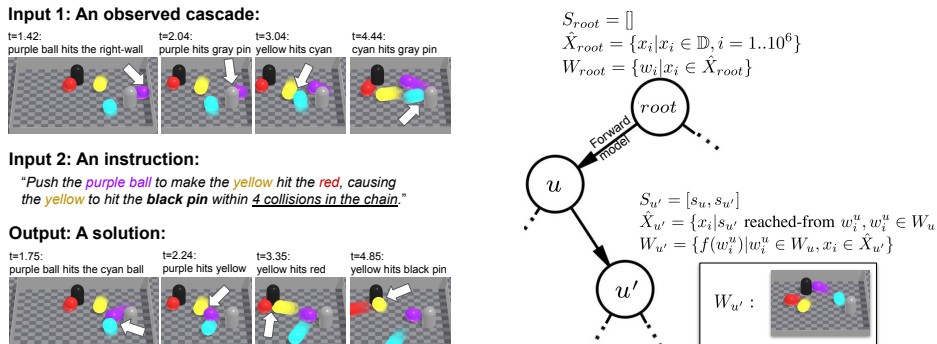

Figure 1: **The *Cascade* learning setup**. **Top-Left**: A set of balls move freely in a confined space. Balls may collide with walls and static pins (grey & black cylinders) and yield an "observed" cascade of events. Arrows highlight each collision. **Middle-Left**: A semantic instruction is given, partially describing an alternative ("counterfactual") cascade of events. **Bottom-Left**: The agent is allowed to intervene and set the (continuous, 2D) initial velocity of the purple ball (the "pivot") to achieve the "counterfactual" cascade of events. Only showing key frames for brevity. See the videos side by side here: https://youtu.be/u1Io-ZWC1Sw (Anonymous). **Right**: Illustration of the event tree data structure.

millions of candidate interventions. Then, one would also need to train a discriminator to decide if the simulation output satisfies the instruction goal, which is hard by itself. Needless to say, searching exhaustively over the space of possible interventions is not feasible. Another approach would be to train a regression model that takes the semantic goal and a representation of the "observed" cascade and outputs the predicted intervention. We describe our experiments with this approach and explain why such fully differentiable approaches struggle in our problem.

Here, we design a representation that focuses on the semantics of the system dynamics (Figure 1). We call it an *Event Tree*. Specifically, we only keep key events where objects interact. In our physical system, these semantic events are collisions of balls, pins, and walls. We use these collisions to build a tree of possible behaviors of the system. In this tree, each node corresponds to a possible collision event. Every node $i$ is connected to its children $i_1, ..., i_k$ which correspond to $k$ realizable future collisions. A path in the tree therefore captures a realizable sequence of collisions. This representation also implies a tessellation of the space of possible interventions. Each intervention (a point in the intervention space) dictates a path in the event tree. Similar interventions often follow the same path.

To build this event tree, we assume that we have a special kind of forward model operating at the level of semantic events. For any given initial condition, we can query the forward model for the next event (e.g., "which objects collide next?"), and predict the outcome of that event (the directions and velocities of objects after they collided). Importantly, every node in the event tree is assigned a value. We learn a function that assigns values to nodes conditioned on the instruction.

Finally, we propose a simple approach for "counterfactual" reasoning that uses the observed cascade. We first find the path in the event tree that corresponds to the observed video. Then, we perform a tree search in the event tree and explore first those paths that split off that observed path.

This paper makes the following contributions: (1) A new learning setup, *Cascade*, where an agent observes a dynamical system and then changes its initial conditions to meet a given semantic goal. (2) A data structure – *Event Tree* – and a principled probabilistic value function for searching efficiently over the space of interventions. (3) A method to transform a tree path into a Directed Acyclic Graph and learn its value function using a Graph Neural Network. (4) A method to use observed cascades to guide the search in the event tree towards a "counterfactual" outcome of the system.

## 2 *Cascade*: A BENCHMARK FOR REASONING AND INTERVENING IN A CASCADING DYNAMICS.

We designed a new environment for evaluating reasoning and intervening in cascading events. The environment combines visual and physical modalities with semantics and action (Figure 1).

**Scenes.** Each scene describes a unique dynamical system. In these systems, several spheres move freely on a frictionless table, colliding with each other and with static pins within a confined four-

walled space (Figure 1). Each episode describes a different scene, which includes tens of collisions. The agent is provided with (1) An "observed" (unperturbed) video of the evolution of this system; and (2) The initial position and velocity of each object.

**Instructions.** An instruction describes (1) A set of target semantic events (collisions) to be fulfilled "*Make the red ball hit the black pin*"; (2) A pivot object to manipulate "*You can push the green*"; and (3) constraints, of two possible types, that encourage the agent to reason about the cascade of events. The first type is a "count" constraint, e.g. "*Within 3 collisions in the chain*". The second type is an "ancestor" constraint, e.g. "*It should include hitting the top wall with the red*".

**The agent's goal.** The agent's objective is to intervene with the scene's initial conditions by setting the velocity vector of the pivot object to reach a set of collisions specified by the instruction. This often requires a highly precise "trick shot", that takes into account detailed reasoning on how downstream events will roll out. Our instruction generation process guarantees that a solution exists, i.e, there are feasible interventions that generate a sequence of events that satisfy the instruction.

**The problem setup.** An agent is given three inputs: An initial condition of the system, an "observed" cascade (e.g., a video) of the system that starts from these initial conditions, and a semantic instruction. The agent is asked to intervene by controlling one "pivot object" in the scene. It is allowed to adjust its velocity (movement direction and speed) to any value in a bounded set of possible interventions $\mathcal{X} \subset \mathbb{R}^d$. Given an instruction $g$, there exists a subset of possible solutions of $\mathcal{X}$ that, when played out in the real-world, will satisfy the goal $g$. The goal is to predict an intervention $x \in \mathcal{X}$ that fulfills the instruction $g$.

At inference time, the agent is only given the instruction and the "observed" cascade, and it should predict how to correctly intervene on the pivot initial conditions.

## 3 METHODS

Our proposed approach focuses on key "semantic" events of the dynamics (e.g., collisions). We build a tree of possible outcomes such that a path in the tree captures a realizable cascade of events. In addition, we learn a function that assigns values to tree nodes conditioned on the instruction.

**The tree of possible futures**: We now describe the main data structure used for representing the search problem - the *Event Tree*. The event tree is designed to provide a searchable data structure for sequences of events that are physically realizable. To make these searches efficient, we represent the system behavior at the *semantic* level. In our concrete physical example, a semantic event $s_i$ is a collision between two object (for example, the yellow ball "*hits*" the right wall). In other systems, these could be any key interactions between system components, like "*manufacturing*" an item in a supply chain. Importantly, in our approach, we require that it is possible to compute the state of the system right after an event.

In our event tree, each node corresponds to a sequence of semantic events. The children of each node correspond to realizable continuations of the event sequence. Namely, all possible events that could happen after the sequence $S_u$. We now formally describe the properties of nodes and how we build the edges of the Event Tree.

**Tree Nodes.** A node $u$ of the event tree represents a sequence of semantic events $S_u \triangleq (s_1, s_2, \ldots s_u)$. Note that there could be many interventions in $\mathcal{X}$ that lead to the same sequence of events $S_u$. As a result, a node also corresponds to the subset of interventions $X_u \subset \mathcal{X}$ that yields the sequence of events $S_u$. We aggregate to a common node all interventions $x \in \mathcal{X}$ that result in the same sequence prefix. The root node describes the dynamical system at $t = 0$ and its sequence of events $S_{root}$ is empty. Its *intervention subset* is $X_{root} = \mathcal{X}$ ( See Figure 1, top). For every node, we also keep the *world-state* $W_u$, which is the set of physical configurations of the system after yielding the sequence $S_u$, for every $x \in X_u$ (Figure 1). Given the world state of a current node, we then ask for the next immediate event (next feasible immediate collision).

**An event-driven forward model.** The goal of the forward model is to uncover which semantic events may occur next. Conceptually, our model takes as input a world-state $w_i$ and outputs the next immediate semantic event. Our forward model $f(\cdot)$ is not a full fledged simulator as it does not propagate time at discrete time steps but solves a set of analytic equations to find the event time, should one occur. However, our model only provides an *approximation* of the dynamics of our test bed environment. These sim-to-sim differences mimic the sim-to-real problem.

**Node Expansion.** Suppose we decide to expand a node $u$. We apply the forward model $f(\cdot)$ to each world state $w_i \in W^u$. We then aggregate all the propagated world states which share the same immediate next semantic event $s'$ to a new node $u'$: The event sequence of the child node $u'$ is $S_{u'} = \text{concat}(S_u, s')$, the corresponding interventions are $X_{u'} = \{x_i | s' \text{ reached-from } w_i\}$ and $W_{u'} = \{f(w_i) | w_i \in W_u, x_i \in X_{u'}\}$.

Expanding the tree can be viewed as a tessellation refinement of the intervention space $\mathcal{X}$. At each step, we pick one cell and split it into multiple cells, where each child cell represents a different event that occurs after a shared sequence of events, represented by the parent cell.

If the tree is fully expanded, it covers all possible futures. However, expanding the whole tree is expansive, because the branching ratio, or the number of events, grows quadratically with the number of objects. In the next subsection, we discuss how we learn a scoring function and use it to guide an efficient tree search.

**Learning the value function**: To find a node that satisfies the goal, we prioritize which node to expand by learning a value function that is conditioned on the instruction $g$. We take a principled probabilistic approach for setting the value function, and set it to the likelihood that the sequence of events of $x \in X_u$ will satisfy the goal $g$, $V(u) = \Pr(x \text{ satisfies } g | g, x \in X_u)$. This probabilistic perspective allows us to take a maximum-likelihood approach at inference time.

**A model for the value function.** The model takes as inputs the instruction $g$ and sequence of events $S_u$ that define the node $u$, and predicts a scalar value. We propose to transform each sequence to a Directed Acyclic Graph (DAG) that captures relations in the cascade of events. A node in this DAG is an event that involves some objects (a collision), and each edge represents a dynamic (moving) object shared by two subsequent events (Figure 2). We chose to use a popular message passing GNN model (1) that maintains learnable node, edge and global graph representations.

**Inference**: Our agent searches the tree for the maximum valued node $u_{MAX}$. Then, it randomly selects an intervention from its intervention subset $x \in X_{u_{MAX}}$. We consider two variants.

**Interventional search:** The agent performs a tree search for the highest valued node. At any given step, the agent stores a sorted list of nodes together with their values, it then picks the highest valued node from this list and expands it. The node children are then added to the list with their predicted values, and the agent resorts the list.

**Counterfactual search:** To improve the interventional search, we to use the *observed* cascade. Consider the case where the sequence of the solution is complex (long sequence) and the *observed* sequence diverges from the solution at a late point. In this case, it is likely that the observed will be informative about the solution. Therefore, we start the search by expanding the nodes along the observed sequence: For each observed node, we add its children to the list described above together with their predicted values. We then continue the search as described by the "Interventional search".

## 4 EXPERIMENTS

We sampled $\sim 46K$ scenes, each includes 4-6 moving balls, 0-2 pins, and 4 walls and up to 5 semantic instructions ($\sim 4.25$ on average). An episode is a pair of a scene and one instruction.

**Compared Methods.** Our full-fledged approach is **ROSETTE (Reasoning On SEmanTic TreEs)**: Search uses the "counterfactual" variant of the tree search (Section 3), by first expanding the nodes along the "observed" sequence. **ROSETTE-IV**: Like ROSETTE, but using "Interventional search" (Section 3) without using the "observed" sequence. **SEQUENTIAL**: Using a sequential representation for a tree chain, instead of a DAG. **Deep**

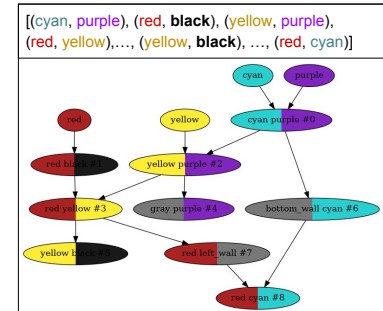

Figure 2: Illustrating how a sequence of events (top) is transformed to a DAG (bottom). It corresponds to the video in Figure 1 bottom.

**Sets regression**: Embedding the instruction and the initial world state to predict a continuous intervention. **Random**: Sample intervention at random from an estimated distribution of ground-truth interventions. **Brute force**: We train a classifier to detect goal satisfaction given a sequence of collisions. Then, search using the event-driven forward model over $10^6$ initial conditions.

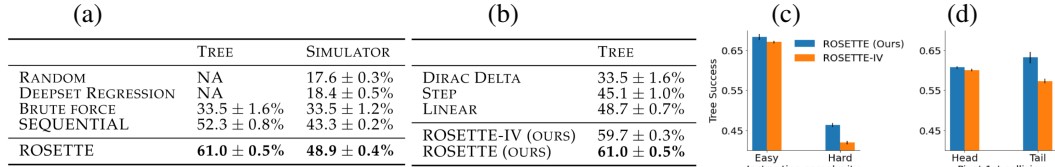

Figure 3: **(a)** Success rate of our approach and baselines. **(b)** Tree success rate for variants of the value function. **(c)** Comparing "Counterfactual search" (ROSETTE) with "Interventional" search (ROSETTE-IV) for easy and hard instructions. **(d)** As in c, but based on the starting point of the pivot in the sequence. *Tail* means that the first pivot collision happens after more than 5 events. ROSETTE performs significantly better, showing improvement of 10.7% and 13% respectively.

**Evaluation metrics.** We consider two metrics. **Simulator success rate**: The success rate when rolling out the predicted intervention using a physical simulator (2). This metric mimics experimenting in the real world. **Tree success rate** : We compare the sequence of semantic events from the selected node in our event tree with the events and constraints specified by the instruction. This metric allows us to evaluate the performance of the value function model and tree search, independently from the errors that may be introduced due to the event-driven forward model.

We measured the tree success rate by conditioning on properties of the instruction and scene. **(1) Condition on instruction complexity:** Instructions with 2 or more constraints are marked as "Hard".**(2) Condition on pivot first collision**: The starting point of the first collision of the pivot in the ground-truth "counterfactual" video. We aggregate the success rate over "Head" cases, where "Pivot 1st collision" $\in \{1 \ldots 5\}$, covering ~80% of the test data, and the remaining "Tail" cases.

**Results**: Table (a) in Figure 3 describes the *Tree* and the *Simulator* success rates of ROSETTE and compared methods. ROSETTE achieves the highest success rate for both the "Tree" success rate (61%) and the "Simulated" success rate (48.9%). When evaluating the brute force approach with same computational budget as ROSETTE, expanding 80 events, it performs largely worse than ROSETTE (33.5%). However, when allowing a $\times 20$ *computational budget*, expanding 4000 nodes (4.5 min/episode vs 13 sec/episode), it reaches a similar success rate as ROSETTE ($Tree = 61.9\%, Simulated = 53.8\%$).

We carried out a set of ablation experiments that quantify (1) The benefits of using the "Counterfactual search" (2) Comparing with various value functions: **(1)** Figure 3(c, d) quantifies the benefit gained by using "Counterfactual" (ROSETTE) over "Interventional" (ROSETTE-IV) search (Section 3). **(2)** Table (b) in Figure 3 shows the advantage of the probabilistic formulation of the value function (ROSETTE-IV), compared to the several heuristics described in Section 4.

**Qualitative Examples**: Here we provide links to qualitative examples we uploaded to YouTube, best viewed in $\times 0.25$ slow motion. We compare ROSETTE successes with ROSETTE-IV failures. link #1, link #2, link #3, link #4, link #5, link #6. ROSETTE followed the observed cascade along the part of the path that was useful to satisfy the instruction. It diverged from the path when necessary, and found a solution when long cascades were essential, while ROSETTE-IV struggled.

## 5 DISCUSSION

We presented a new learning setup, called *Cascade*, where an agent observes a cascade of events in a dynamical system and is asked to intervene and changes its initial conditions to meet a given semantic goal. The problem we try to solve is inherently not a standard planning problem. An action is taken essentially only once and there is no way to change the ensuing cascade of events using additional actions ("Fire and forget"). This problem is a complex search problem which one can conceivably try to solve as such using brute-force methods or using some metaheuristic approaches.

We use an event tree representation and a principled probabilistic value function for searching efficiently over the space of interventions. In addition, we show that "hot starting" the tree search using the observed cascade improves the success rate.

**Related work** CLEVRER, CRAFT, CATER, IntPhys and CoPhy (3; 4; 5; 6; 7) are video understanding benchmarks, exploring reasoning over observed temporal and causal structures. They focus on understanding, question answering and tracking rather than taking an action as *Cascade* do. (8; 9; 10; 11), learn physical forward models that focus on object interactions, using a fixed time step, rather than an adaptive time step like we do.

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
