# OpenReview forum: "Learning to reason about and to act on physical cascading events"
_ICLR.cc/2022/Workshop/OSC — ICLR2022 OSC  Poster_

### Official Review · Reviewer_nXtJ · 2022-03-14

**Rating:** 2
**Confidence:** 3

**Review:**

Summary: The authors introduce a new problem setup where the agent observes a trajectory and is given a text description of a target trajectory. The agent must predict new initial conditions (i.e. an intervention) for a given pivot object such that when the dynamics are rolled out, the resulting trajectory conforms to the input text description. The authors propose to do this prediction by constructing an event-based tree and then performing a symbolic search on this tree.

Pros:
1. The idea of the event tree is interesting and novel.
2. The experiment results show the clear benefit of using the event tree in comparison to feed-forward prediction using DeepSets. It is also more computationally efficient than the brute force method.
3. Analysis shows the benefit of using a GNN for learning the value model.

Cons:
1. Description of the forward model is limited and it is unclear if it is learned from data.
2. What does the forward model output in more formal terms?

Conclusion: The paper is interesting and relevant for the workshop.

---

### Official Review · Reviewer_bvVj · 2022-03-16

**Rating:** 3
**Confidence:** 3

**Review:**

Summary: the authors propose a novel learning setup, Cascade, that tasks the agent to intervene on the initial conditions of the scene to achieve a counterfactual outcome, given an observed scene. The agent is provided with an instruction that gives the agent a counterfactual goal to achieve, provides a hint on which object to intervene on, and provides constraints the agent must satisfy. This is challenging problem and the authors provide a method for solving this task.

Strengths:
- A novel data structure, the Event Tree, that make searching through possible futures efficient
- A value function for focusing the search
- Leveraging observed data to inform the search

Weaknesses:
- the proposed approach requires much domain knowledge of the underlying system in order to build the event tree. Such domain knowledge may not necessarily be available, or expensive to get, in applications.

Overall, however, I see this as a novel and interesting benchmark and very much relevant to the workshop theme.

---

### Decision · Program_Chairs · 2022-03-24

**Decision:**

Accept (Poster)

**Comment:**

The reviewers agree the paper should be accepted at the workshop. Congratulations!

The authors are encouraged to take the points raised by reviewer nXtJ into account when preparing the camera-ready version.